# Nonoxidative Coupling of Methane to Produce C_2_ Hydrocarbons on FLPs of an Albite Surface

**DOI:** 10.3390/molecules28031037

**Published:** 2023-01-19

**Authors:** Yannan Zhou, Ye Chen, Xuegang Luo, Xin Wang

**Affiliations:** 1Research Center of Laser Fusion, China Academy of Engineering Physics, Mianyang 621900, China; 2Qinghai Institute of Salt Lakes, Chinese Academy of Sciences, Xining 810008, China; 3Engineering Resarch Center of Biomass Materials, Ministry of Education, Mianyang 621010, China

**Keywords:** albite, frustrated Lewis pairs, methane, nonoxidative coupling on surface, doping modification of catalyst

## Abstract

The characteristics of active sites on the surface of albite were theoretically analyzed by density functional theory, and the activation of the C-H bond of methane using an albite catalyst and the reaction mechanism of preparing C_2_ hydrocarbons by nonoxidative coupling were studied. There are two frustrated Lewis pairs (FLPs) on the (001) and (010) surfaces of albite; they can dissociate H_2_ under mild conditions and show high activity for the activation of methane C-H bonds. CH_4_ molecules can undergo direct dissociative adsorption on the (010) surface, whereas a 50.07 kJ/mol activation barrier is needed on the (001) surface. The prepared albite catalyst has a double combination function of the (001) and (010) surfaces; these surfaces produce a spillover phenomenon in the process of CH_4_ activation reactions, where CH_3_ overflows from the (001) surface with CH_3_ adsorbed on the (010) surface to achieve nonoxidative high efficiently C-C coupling with an activation energy of 18.51 kJ/mol. At the same time, this spillover phenomenon inhibits deep dehydrogenation, which is conducive to the selectivity of the C_2_ hydrocarbons. The experimental results confirm that the selectivity of the C_2_ hydrocarbons is maintained above 99% in the temperature range of 873 K to 1173 K.

## 1. Introduction

Direct conversion of methane (DMC) to high value-added chemicals (such as ethylene and other compounds) is an effective way to achieve its efficient utilization, which plays a vital role in the economy and environment. Methane is the most stable small molecule in nature, due to its stable tetrahedral symmetric structure, which causes its efficient activation and direct conversion under mild conditions to be extremely challenging. After decades of continuous efforts, researchers have proposed many ways to produce DMC, among which the development of oxidative coupling, nonoxidative dehydrogenation and selective oxidation methods has greatly promoted research on DMC. In recent years, the high utilization rate and high selectivity of the active species of single-atom catalysts (SACs) have attracted extensive attention from researchers [1,2,3,4]. Because of their unique catalytic performance in DMC reactions, SACs have been used in various reactions, and although they have many advantages, they still face many challenges, due to their high surface energy and instability of single atoms along with limitations of characterization methods for the nanoscale to single-atom scale species, catalyst design and preparation, etc. Therefore, the goal is to find catalysts with high catalytic performance, simple preparation and good stability.

Since 2006, Professor W. Stephan has found that homogeneous frustrated Lewis pair (FLP) catalysts could activate the H_2_ molecule [5] under mild conditions, and the activation potential of FLP for small molecules in homogeneous fields has constantly been of great interest to researchers. However, the application of homogeneous FLPs is hindered by defects such as product separation and the recycling of catalysts; some researchers have gradually begun focusing on developing heterogeneous FLP catalysts on a solid surface, aiming to make a breakthrough in DMC. At present, all-solid FLPs constructed from a variety of materials, such as doped graphene [6], B/Al-doped phosphorenes [7], and hydroxylated indium oxide [8], have shown good activity performance for H_2_ activation and hydrogenation reactions. Since research in this area has just began, there have been relatively few studies on the nonoxidative activation of CH_4_. Metal oxides such as the FLP of γ-Al_2_O_3_ (110) and CeO_2_(110) have a remarkable ability to activate the methane C-H bond [9,10], and nonoxidative C-C coupling can be performed on the surface of CeO_2_. Recently, researchers have extended the concept of FLP to multistage porous Ti-based and zeolite molecular sieve catalytic systems [11,12] and made important progress in the research of photocatalytic nonoxidative coupling of methane and dehydrogenation of hydrocarbons above C_3_. To date, the reference research methods and concepts are very limited. However, solid FLPs show the potential to activate methane in catalytic reactions, which attract further research and exploration to achieve different goals in FLP research.

Albite (Ab) is a type of natural mineral that is abundant, cheap, and easy to obtain. It not only has good thermal stability and chemical stability but also has channel structure characteristics similar to those of zeolites. The difference is that the pore diameter of the channel is below 3 Å and belongs to the ultra-microchannel; due to this difference, researchers have not pursued long-term studies using albite. In an experimental study, we found that albite catalysts have a certain activation effect on methane using nonoxidative conditions, and the C_2_ hydrocarbon selectivity is above 99% [13]. Using density functional theory (DFT), it is found that FLPs exist on the surface of albite, and we speculate that they activate methane according to the mechanism of FLPs. Therefore, it is necessary to further study the nature and trends of methane activation using albite catalysts in theory, which is of great significance to expand the application of ultra-microchannel mineral materials.

In this study, initially, the FLP active sites on the (001) and (010) surfaces of albite were theoretically analyzed, and the calculation confirmed that they had the characteristics of FLPs. Then, we studied in detail the activation effects and conversion behaviors of surface FLPs to CH_4_ and determined the mechanism showing its high selectivity to C_2_ hydrocarbons and its non-easy production in carbon deposits. Finally, the static electric field of FLPs was regulated by doping modification, and the influence on CH_4_ conversion was analyzed and discussed to obtain a newfound understanding of the application of albite and provide a novel concept and method for the nonoxidative coupling of methane by solid FLPs.

## 2. Results and Discussion

### 2.1. Theoretical Analysis of FLP Sites on Albite Surfaces

The basic structural units of albite (NaAlSi_3_O_8_) are [SiO_4_] and [AlO_4_] tetrahedrons, and each oxygen atom in the skeleton is shared with two adjacent tetrahedrons. Because of the charge imbalance caused by Al replacing Si, a Na^+^ cation with balanced charge is introduced into the structure and is located in the pores of the tetrahedral skeletal structure. Albite minerals have two common exposed surfaces, (001) and (010), in a natural or grinding external force situation [14,15], and the structures are shown in Figure 1.

Initially, we analyzed the active sites in the two surface structures of albite according to Lewis acid–base theory. The tetrahedral central atoms in albite structures Al and Si have four sp^3^ hybrid orbitals, which are bonded to four O atoms on the (001) surface. [AlO_3_] lacking one oxygen is exposed on the surface layer. Al is in the coordination unsaturated state and has an empty orbital, which can accept electron pairs; Al acts as a Lewis acid (LA). The oxygen atom exposed on the surface has an electron rich attribute and acts as a Lewis base (LB). [SiO_3_] lacking oxygen is exposed on the (010) surface. Si is in the coordination unsaturated state, its valence orbit can accept electrons, and it acts as a LA; similarly, the oxygen atom exposed to the surface acts as a LB. Both LAs and LBs on the two surfaces are distributed on the skeleton of [AlO_4_] and [SiO_4_]. The formation of solid frustrated Lewis acid–base pairs requires not only independent LAs and LBs but also appropriate sterically encumbered matching pairs. Albite is a mineral with an ultra-microchannel structure, and the pore diameter parallel to the channel is mainly below 0.3 nm [16,17,18]. Due to its undulating and uneven surface, many micropores are formed that are not parallel to the crystal surface; that is, the Lewis acid and base sites distributed on the surface skeleton of albite are not in the same horizontal plane. On the (001) surface, there were two sterically encumbered combinations: one group was located on the channel skeleton, with a distance of 4.21 Å, and the other group was located on the pore skeleton, with a distance of 4.39 Å. On the (010) surface, there were also two combinations: the distance on the channel skeleton was 5.10 Å, and the distance on the pore skeleton was 4.20 Å. Using the viewpoint that a distance between Lewis acid–base sites of approximately 4 Å is appropriate [12,19,20], the Lewis acid–base pairs on the channel skeleton of the (001) face and the pore skeleton of the (010) face are matched; therefore, there are FLPs on both surfaces of albite.

Second, to further verify the characteristics of FLPs on the surface of albite, we calculated the reaction barrier for activating H_2_ (see Figure 2). The results showed that FLPs on both surfaces could enable H_2_ heterolysis. The dissociation activation energy of the (001) surface was 36.48 kJ/mol, and that of the (010) surface was 1.01 kJ/mol. Heterolysis fragments of H^−^ and H^+^ were adsorbed on the LA and LB, respectively. Erker et al. studied the activation mechanism of small molecules H_2_ by FLPs and thought that the activation of small molecules by FLPs was based on the polarization of the electrostatic field between the electron receptor and donor acid–base pair [21,22]. Using a Lewis acid–base electrostatic field, an activity region would be formed, and the energy barrier in the reaction process was generated in the preparatory entry stage of small molecules. Once small molecules entered the active region of FLPs, the reaction energy barrier disappeared, and a stable dissociated product was finally formed [23]. We also found in the calculation that when H_2_ is at a suitable distance from the Lewis acid–base pair on the two surfaces, direct dissociation adsorption occurs. When H_2_ enters the active region formed by LA and LB, it is directly dissociated. The transition-state (TS) structure of H_2_ dissociation (Figure 2) shows that the H_2_ molecule has not been dissociated; in fact, the activation energies of 36.48 kJ/mol and 1.01 kJ/mol are required for H_2_ to enter the activity region from outside. The FLPs on the two surfaces have a strong activation effect on H_2_, showing evident FLP characteristics. Thus, albite is a natural mineral with an ultra-microchannel structure, and its surface coordination unsaturated Si, Al, and O atoms are distributed on the skeleton formed by [AlO_4_] and [SiO_4_], which can form mutually independent LAs and LBs. Its ultra-microchannel structure causes the LAs and LBs to form matching subnanoscale sterically encumbered minerals, thus forming FLPs.

### 2.2. Effect on C-H Bond Activation of Methane

To deeply understand the adsorption and activation mechanism of CH_4_ on the FLPs of the albite surface, we initially calculated the adsorption energies of CH_4_, CH_3_, H and CH_3_/H on the two surfaces (see Table 1). Using the table, the most stable CH_4_ is adsorbed on the LA (Al,Si), and the most stable adsorption of the co-adsorption system of CH_4_ dissociated species CH_3_/H is located on the LA (Al,Si)/LB (O). Additionally, the adsorption energies of CH_4_ and dissociated species CH_3_, H on the two surfaces follow the trend of CH_4_ < CH_3_ < H. The adsorption of CH_3_ and H on the (010) surface is stronger than that on the (001) surface, and their adsorption energies are 150.09 kJ/mol and 102.4 kJ/mol higher, respectively. The adsorption of CH_4_ on the two surfaces had a similar phenomenon to that of H_2_, showing different adsorption between the CH_4_ and LA (Al,Si) distances and degrees of polarization, as shown in Figure 3. On the (001) surface, when CH_4_ is far from the LA (Al) site (d_C-Al_ > 3.8 Å), the adsorption energy is positive, indicating instability (see Figure 3a). When CH_4_ is close to the LA (Al) site (d_C-Al_ < 3.8 Å), CH_4_ is stably adsorbed at a position of 2.34 Å from C to Al (see Figure 3b). The adsorption energy is −27.86 kJ/mol, generating a dipole moment of 0.13 D, indicating that CH_4_ has been polarized; the C-H bond is weakened and stretched to 1.11 Å. On the (010) surface, when CH_4_ is far from the LA (Si) site (d_C-Si_ > 3.9 Å), the adsorption of CH_4_ on the surface is weak (see Figure 3c). When CH_4_ is close to the LA (Si) site (d_C-Si_ < 3.9 Å), heterolysis occurs directly, CH_3_ is adsorbed on the LA (Si), and H is adsorbed on the LB (O), as shown in Figure 3d. This is completely consistent with the phenomenon described by Grimme S. et al., in that “once the molecule enters the hole of FLPs, the reaction proceeds without a barrier” [23]; that is, CH_4_ can undergo direct dissociative adsorption in the active area of the Lewis acid–base pairs on the (010) surface.

To analyze the micro-mechanism of its dissociative adsorption process, we observed the dynamic change in the (010) surface adsorption configuration, as shown in Figure 4, and the corresponding Hirshfeld charge distribution and CH_4_ dipole moments were calculated (see Table 2). Interestingly, there are three stages in the dissociative adsorption process of CH_4_. In the first stage (a–d), as CH_4_ gradually approaches the LA (Si), the LA (Si) is in a state of acquiring electrons, CH_4_ is in a state of donating electrons, and the LB (O) has no electron transfer; the CH_4_ dipole moment gradually increases, and CH_4_ is in a state of physical adsorption. In the second stage (e-f), when the distance between C and Si is 2.23 Å, CH_4_ chemisorbs with the LA (Si). At this time, the LA (Si) still maintains a state of acquiring electrons, the LB (O) begins to enter a state of donating electrons, and the dipole moment of CH_4_ rapidly increases from 0.24 D to 0.57 D. The C-H bond closest to the LB (O) is stretched to 1.23 Å. In the third stage (g-h), when the C-H bond is stretched to 1.34 Å, the C-H bond is broken, and the H-LB (O) bond forms at the same time. After reaching a stable state, CH_3_ and H are adsorbed on the LA (Si) and the LB (O), respectively, with charges of −0.07 e and +0.18 e. Based on the above reaction details, when CH_4_ is adsorbed on the LA (Si) on the albite (010) surface, LA (Si) can obtain electrons and has strong electrophilicity. Therefore, it initially attacks the electron-enriched C atom in CH_4_, causing the electrons from the CH_4_ bonding orbital σ to flow to the LA (Si) on the surface to form a Si-CH_4_ intermediate; the C-H bond of CH_4_ is weakened, which then promotes the nucleophilic attack of the LB (O) and provides electrons to the antibonding orbital σ* of the C-H bond of CH_4_ until it ruptures. In the process of CH_4_ activation, the LA, as an electrophilic reagent, plays an important role in the polarization of CH_4_; a greater degree of polarization of CH_4_ correlates to a stronger reaction activity. The LB, as a nucleophilic reagent, can obtain protons, and they synergistically complete the cleavage of CH_4_. Since the electrophilicity of the LA on the (001) surface is weaker than that on the (010) surface, its polarization to CH_4_ is relatively weak, and CH_4_ cannot be directly cleaved.

Second, the reaction barriers of CH_4_ activation by two surface FLPs were calculated (see Figure 5); both surfaces show good activity. The reaction heat of CH_4_ dissociation on the (001) surface is −71.72 kJ/mol, and an activation energy of 50.07 kJ/mol needs to be overcome. The reaction heat of CH_4_ dissociation on the (010) surface is −292.61 kJ/mol, and the activation energy is 2.63 kJ/mol. From the two surface TS structures, CH_4_ on the (001) surface has been dissociated, while CH_4_ on the (010) surface has not been dissociated. CH_4_ was polarized and contained the activation energy required for methane to enter the activity region from outside. Once CH_4_ enters the activity region, heterolysis occurs according to the direct dissociation mechanism.

From a thermodynamic point of view, the dissociation reaction heat of the (010) surface is far greater than that of the (001) surface, indicating that the bonding effect between dissociated fragments CH_3_ and H with acid–base sites of the (010) surface is stronger than that of the (001) surface. From a kinetic point of view, the activation energy of CH_4_ on the (010) surface is much lower than that on the (001) surface. Although the dissociation activation energy of CH_4_ on the (010) surface is the lowest, the strong bonding effect between CH_3_/H and the surface means that desorption requires a higher desorption energy, which is in accordance with the Brónsted Evans Polanyi rule [24].

### 2.3. Effect on the Methyl C-C Coupling Mode

The albite catalyst is prepared by high-temperature calcination and grinding using an external force to physically mix and fully expose the two common exposed surfaces, (001) and (010). In fact, it is a double combination catalyst with (001) and (010) coexisting, and they participate in the catalytic reaction of methane at the same time. Figure 6 shows the catalytic conversion path of methane on the albite catalyst; when methane enters the catalyst surface from the gas phase, the FLPs on both surfaces cause methane heterolysis according to the same activation mechanism, and the dissociated fragments CH_3_ and H are located on LA and LB sites on both surfaces, respectively. Since the bonding effect with CH_3_ and H adsorbed on the (001) surface is much weaker than that on the (010) surface, they preferentially desorb during temperature rise desorption, causing a spillover phenomenon; at the same time, because the Lewis acid site Al on the (001) surface has an empty orbital and CH_3_^−^ has a lone electron pair, after the two bond, that is, CH_3_^−^ adsorbed on (001) surface can neither give nor gain electrons, C-C coupling does not occur on (001) surface. Similarly, the spilled H^+^ cannot gain electrons from Lewis acid site Al in (001) surface to form H^−^. Therefore, the coupling reaction of CH_3_ and H overflowers on (001) surface can only occur on (010) surface, that is, CH_3_ and H preferentially desorbed become overflowers and diffuse to the (010) surface with an increase in the concentration gradient. In the overflow process, since the bonding effect of CH_3_ on the (001) surface is much smaller than that of H, CH_3_ desorbs first.

Figure 7 shows the desorption energy of CH_3_ spillover and H spillover on the (001) surface and the reaction energy barrier on the (010) surface. The desorption of CH_3_ requires an activation energy of 306.37 kJ/mol, and the diffusion to the (010) surface is coupled with the adsorbed methyl group through the E-R mechanism. The formation of C_2_H_6_ is an exothermic process and only requires an activation energy of 18.51 kJ/mol, indicating that C-C coupling has a strong bonding effect and shows good activity. CH_3_ is E-R coupled on the surface, which is advantageous both in thermodynamics and kinetics, and part of the C_2_H_6_ generated by coupling continues to dehydrogenate and then generates C_2_H_4_. The above results are consistent with the experimental results. At 1073 K, the analysis of the relationship between CH_4_ conversion, C_2_ hydrocarbon selectivity and space velocity (GHSV) shows that CH_4_ conversion decreases with increasing GHSV; CH_4_ activation is fully achieved on the surface FLPs because a higher GHSV corresponds to a shorter contact time and lower CH_4_ conversion rate. The phenomenon of increasing C_2_H_6_ and decreasing C_2_H_4_ with increasing GHSV further confirms that the coupling of CH_3_ to produce C_2_H_6_ occurs on the surface rather than in the gas phase. C_2_H_4_ is the product of dehydrogenation of C_2_H_6_ in the gas phase [13]. When H overflows and diffuses to the (010) surface, and since the electronegativity of the H atom (2.1) is greater than that of the Si atom (1.8), H spillover can easily acquire unpaired electrons from the LA (Si) to form H^−^. This reaction process produces a strong exothermic reaction with a reaction heat of −431.12 kJ/mol and an activation energy of 24.69 kJ/mol. After that, H adsorbed on the LA (Si) reacts with proton H adsorbed on the LB (O) to generate H_2_ through the L-H mechanism; the required activation energy is 306.08 kJ/mol, and the closed cycle of methane conversion is completed. Since the desorption energy required for H spillover on the (001) surface (387.36 kJ/mol) is the highest during the whole reaction process, this is the rate-determining step in the whole cycle process.

This spillover phenomenon not only enables highly efficient coupling of methyl groups but also plays two important roles: (1) to remove CH_3_ and H, which are firmly adsorbed on the surface of (010), to regenerate the acid–base sites and (2) to avoid deep dehydrogenation, which is conducive to the selectivity of C_2_ hydrocarbons. Because the heterolytic activation of CH_4_ by FLPs on the surface of albite is a synergistic catalytic mechanism, the dehydrogenation activation of methane can only be achieved using the joint action of the LAs and LBs. After the Lewis acid–base sites are regenerated, the methane conversion cycle reaction can continue; the desorption order of the dissociated fragments adsorbed on the Lewis acid–base sites directly affects the reaction direction of conversion. When the LB site preferentially regenerates, the carbon species adsorbed on the LA site undergo dehydrogenation, whereas when the carbon species adsorbed on the LA site preferentially desorb, the occurrence of deep dehydrogenation is avoided. Since the bonding effect between the (001) face of CH_3_ and the surface is much smaller than that of H, the preferential desorption of CH_3_ avoids further dehydrogenation. At the same time, CH_3_ preferentially overflowed is coupled on the (010) surface to form ethane, which also avoids further dehydrogenation of CH_3_ adsorbed on the (010) surface. This reasonably explains the experimental results that the selectivity of C_2_ hydrocarbons is maintained across up to 99% and there are zero carbon deposits in the temperature range of 873~1173 K.

### 2.4. Effect of Doping Modification on CH_4_ Conversion

From the above analysis, the albite catalyst shows good catalytic ability for methane C-H bond activation and methyl C-C coupling, has high selectivity for C_2_ hydrocarbons and does not easily undergo carbon deposition. However, we also find that the conversion of CH_4_ is relatively low, and it increases with increasing temperature. This phenomenon shows that the conversion of CH_4_ is closely related to temperature. Since the FLP catalytic reaction is generally a synergistic mechanism, the regeneration of acid–base sites is crucial to the cyclic reaction. For albite catalysts, the desorption of dissociated fragment H on the (001) surface is the rate-determining step of the cyclic catalytic reaction. Under nonoxygen conditions, temperature is the main factor affecting the desorption of H, and higher desorption energy can lead to lower conversion. Therefore, on the basis of a lower desorption energy of CH_3_ than that of H, a reduction in the desorption energy of H helps to improve methane conversion. In the albite structure, the Na^+^ in the pores mainly balances the residual negative charges on the skeleton and affects the electrostatic field formed by the LA and LB. According to the principle of field strength superposition, doping high valence metal ions instead of Na^+^ plays a role in regulating the electrostatic field of FLPs. In this experiment, we mixed the prepared albite catalyst with anhydrous lead chloride in different proportions, calcined it at 773 K, and formed a Pb^2+^-doped Pb/Ab catalyst through ion exchange. To investigate the influence of Pb doping on the CH_4_ conversion rate, we calculated the desorption energy of H and CH_3_ before and after (001) surface doping, and the calculated results were analyzed and compared with the experimental values of CH_4_ conversion and C_2_ hydrocarbon selectivity before and after doping at 1073 K (see Figure 8). The desorption energy of CH_3_ after doping is still less than that of H, and the activation barrier of CH_4_ is slightly increased, increasing from 50.07 kJ/mol to 59.21 kJ/mol; this result means that doping has less effect on the catalytic activity of CH_4_, and the desorption energy of H is reduced from 387.36 kJ/mol to 337.46 kJ/mol, showing a decrease of 49.9 kJ/mol. Corresponding to the experimental results at 1073 K, the conversion of methane after doping was promoted by 2.45 times from 3.32% to 8.12%, and the selectivity remained above 99%. After doping, reducing the bonding effect of the (001) surface H has an evident effect on improving the conversion of CH_4_. Therefore, we believe that a nonreduction (or very low reduction) in the activity and a smaller desorption energy of the CH_3_ spillover than that of the H spillover using modification maximumly reduces the bonding effect between H and the surface to greatly improve the conversion rate. The above are our preliminary research results, and the application of albite catalysts requires further in-depth research.

## 3. Calculation Method

In this paper, the DMol^3^ module [25] of MS19.1 software was used for simulation, and DFT was used for calculation. The exchange correlation potential is described by the PBE (Perdew, Burke and Enzerhof) function under the generalized gradient approximation GGA [26]. The electron wave function used the dual atomic orbital plus the polarization function DNP [25] as the basis set, and the truncation radius was 5.2 Å. The convergence tolerance of energy, gradient and displacement were 1.0 × 10^−5^ Ha, 0.002 Ha/Å and 0.005 Å, respectively. In the self-consistent functional (SCF) calculation, the convergence criterion was 1.0 × 10^−6^ Ha, and the smearing value was set to 0.005 Ha. The transition state was determined by the complete LST/QST [27] and nudged elastic band method (NEB) [28]. The atomic charge was calculated by Hirshfeld population analysis.

Albite crystals belong to the triclinic system, space group C_1_, and the lattice constants are a = 0.8137 nm, b = 1.2787 nm, c = 0.7157 nm, α = 94.245°, β = 116.605°, and γ = 87.809° [29]. In this paper, surface models of periodic NaAlSi_3_O_8_ (001) and (010) were constructed as research objects. Both the (001) surface and (010) surface supercell models were 2×1×1; this size ensured that there was no interaction between adsorbed molecules. A vacuum layer of 18 Å ensured that there was no interaction between adjacent albite layers in the *z*-axis direction. The k point parameters of the (001) surface and (010) surface were 2 × 2 × 1 and 3 × 3 × 2, respectively. The upper part and the adsorbate relaxed, while the remaining atoms were fixed in their original crystal positions. The doping calculation used Pb atoms to replace Na atoms in NaAlSi_3_O_8_ pores, and the DFT semicore pseudopotentials (DSPP) core treatment [30] was applied.

## 4. Conclusions

The effect of albite catalyst FLPs on the nonoxidative coupling of CH_4_ was experimentally studied by DFT. The results show that the LA and LB formed on the (001) and (010) surfaces are sterically encumbered by approximately 4 Å due to albite having a unique ultra-microchannel structure and showing significant FLP characteristics. The activity region formed by the electrostatic field of Lewis acid–base pairs could enable H_2_ heterolysis. Additionally, it has remarkable activation ability for methane C-H bonds, causes direct dissociative adsorption of CH_4_ molecules in the active region of the (010) surface, and requires only a 50.07 kJ/mol energy barrier for the dissociation on the (001) surface. The albite catalyst prepared by sintering and grinding is a catalyst with coexisting (001) and (010) surfaces. The bonding effect of the dissociated species CH_3_ with the two surfaces exhibits quite a larger difference; therefore, a spillover effect is triggered during temperature rise desorption. This spillover effect enables CH_3_ to achieve highly efficient C-C coupling under nonoxidative conditions according to the E-R mechanism and can avoid deep dehydrogenation, inhibit carbon deposits, and maintain high selectivity of the C_2_ hydrocarbons. The desorption of H spillover on the (001) surface is a key step in the CH_4_ conversion cycle reaction, and its high desorption energy is the main reason that affects the conversion rate. By doping high valence metal ions, the nucleophilic and electrophilic properties of FLPs can be adjusted, the desorption energy of H can be reduced, the cycle reaction of methane conversion can be accelerated, and the conversion rate can be effectively doubly promoted. Albite, as a natural mineral with an ultra-microchannel structure, is not only abundant, inexpensive, and easy to obtain but also has good chemical and thermal stability. Moreover, the catalyst is simple to prepare, easy to recover and reuse, showing excellent properties for activation of the C-H bond of methane and C-C coupling, as well as high selectivity for C_2_ hydrocarbons. This study not only provides a newfound understanding of mineral materials with ultra-microchannel structures but also provides a novel concept and reference for the further use of ultra-microchannel materials to construct new solid FLP catalysts and for the nonoxidative conversion of methane.

## Figures and Tables

**Figure 1 molecules-28-01037-f001:**
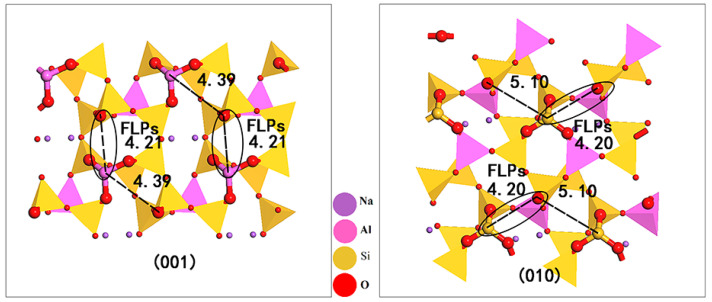
Schematic diagram of the albite (001) and (010) surface model. The circle indicates the FLPs position and the dotted lines indicates the distance between Lewis acid and Lewis base.

**Figure 2 molecules-28-01037-f002:**
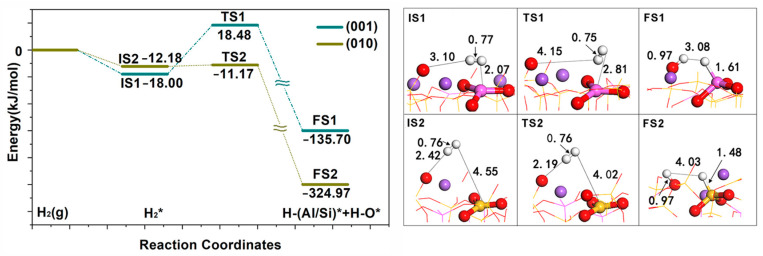
Geometric structures of the transition state intermediates of dissociation reactions of H_2_ on the FLP of NaAlSi_3_O_8_ (001) and (010) surfaces. * indicates adsorption state, (g) indicates in gas phase state.

**Figure 3 molecules-28-01037-f003:**
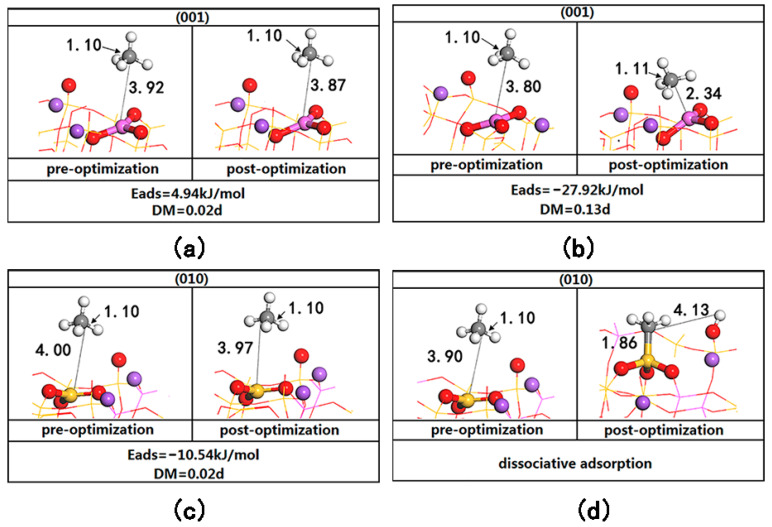
Adsorption configuration of CH_4_ before and after (001) and (010) surface optimization. (**a**) is the adsorption configuration before and after optimization when (001) surface d_C-Al_ = 3.92 Å. (**b**) is the adsorption configuration before and after optimization when (001) surface d_C-Al_ = 3.80 Å. (**c**) is the adsorption configuration before and after optimization when (010) surface d_C-Si_ = 4.00 Å. (**d**) is the adsorption configuration before and after optimization when (010) surface d_C-Si_ = 3.90 Å.

**Figure 4 molecules-28-01037-f004:**
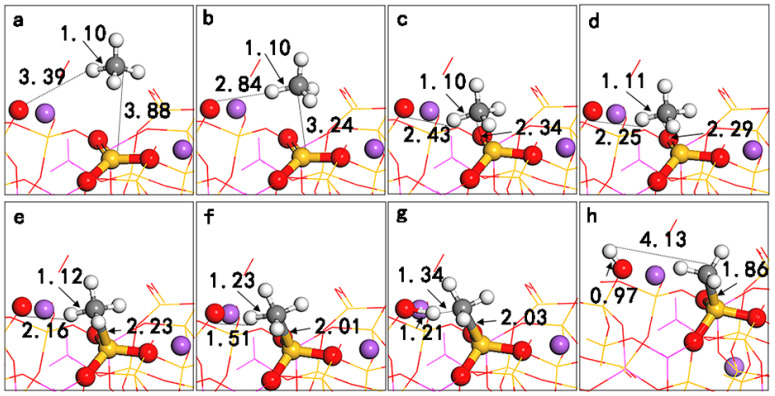
Surface dissociative adsorption process diagram of CH_4_ on NaAlSi_3_O_8_ (010) (**a**–**h**).

**Figure 5 molecules-28-01037-f005:**
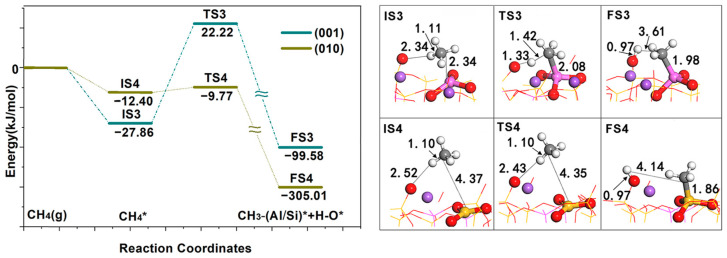
Geometric structures of the transition state intermediates of the activation reactions of CH_4_ on the FLP of NaAlSi_3_O_8_ (010) and (001) surfaces. * indicates adsorption state, (g) indicates in gas phase state.

**Figure 6 molecules-28-01037-f006:**
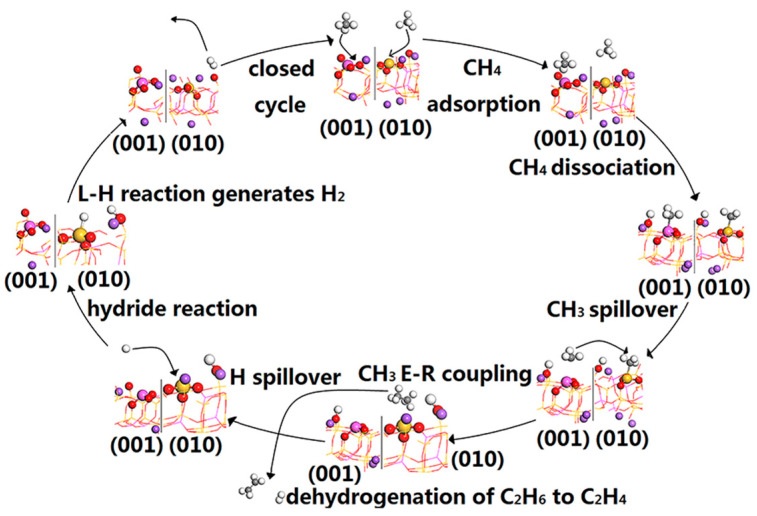
Schematic diagram of the path for activation and conversion of CH_4_ by albite catalyst.

**Figure 7 molecules-28-01037-f007:**
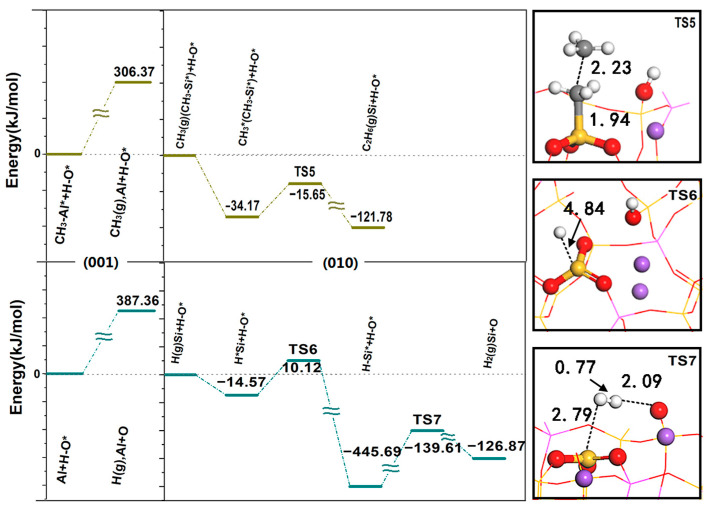
Diagram of desorption energy of CH_3_ spillover and H spillover on the (001) surface and potential energy of reaction on the (010) surface. Corresponding structures of the transition state intermediates in the reaction path calculation. * indicates adsorption state, (g) indicates in gas phase state.

**Figure 8 molecules-28-01037-f008:**
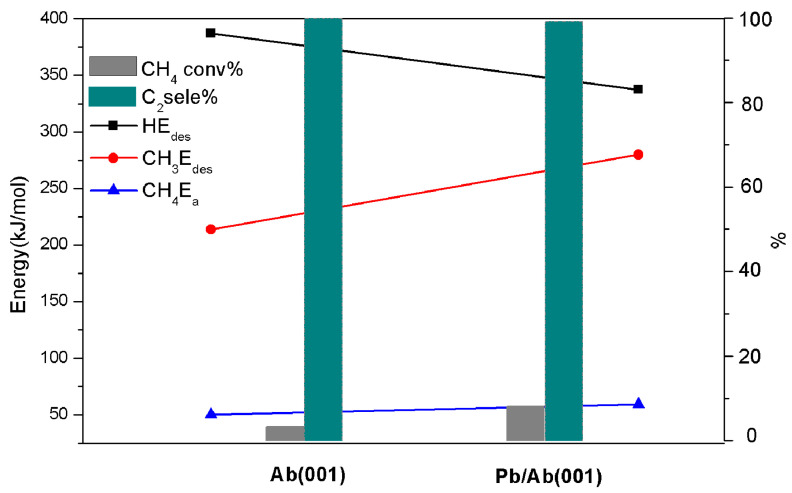
Effect of desorption energy of CH_3_ and H, activation energy of CH_4_ dissociation on CH_4_ conversion and C_2_ selectivity (1073 K) before and after doping on the (001) surface.

**Table 1 molecules-28-01037-t001:** CH_4_, CH_3_, H, and CH_3_/H adsorption energy of the Ab surface (kJ/mol).

Adsorbate	(001)		(010)	
	Sites	Adsorption Energy	Sites	Adsorption Energy
CH_4_	LA	−27.86	LA	−12.4
	LB	−9.31	LB	−5.85
CH_3_	LA	−252.83	LA	−402.92
	LB	−273	LB	−373.29
H	LA	−281.9	LA	−416
	LB	−394.44	LB	−496.84
CH_3_/H	LA/LB	−723.37	LA/LB	−924.25
	LB/LA	−647.39	LB/LA	−817.08

**Table 2 molecules-28-01037-t002:** Hirshfeld charge distribution and dipole moments of the CH_4_ dissociative adsorption process on the NaAlSi_3_O_8_ (010) surface.

State	Hirshfeld Charge Distribution (e)					CH_4_ di-Moments (Deby)
	LA(Si)	LB(O)	CH_4_	CH_3_	H	
a	0.71	−0.45	0.01	-	-	0.01
b	0.70	−0.45	0.03	-	-	0.04
c	0.61	−0.46	0.22	-	-	0.16
d	0.61	−0.45	0.23	-	-	0.19
e	0.60	−0.45	0.24	-	-	0.24
f	0.56	−0.40	0.21	-	-	0.57
g	0.55	−0.33	-	0.05	0.08	-
h	0.50	−0.26	-	−0.07	0.18	-

## Data Availability

Not applicable.

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
