# Peer review of "Nonoxidative Coupling of Methane to Produce C2 Hydrocarbons on FLPs of an Albite Surface"

_molecules, 2023, doi:10.3390/molecules28031037_

Round 1

Reviewer 1 Report

The entitled manuscript “Nonoxidative coupling of methane to produce C2 hydrocarbons 2 on FLPs of an albite surface” by Zhou et al, were reviewed thoroughly. The manuscript is very well-written and the authors paid attention and efforts to market their work in this perfect shape. I strongly recommend the manuscript to be published in Molecules; however, some points were found and could be taken into considerations.

Line 11: albite were experimentally analyzed by density functional theory, and the … I think the authors meant computationally/theoretically instead of experimentally.

Lines 117: and the structure is shown in Figure 1. Please change to structures are shown ….

Line 221: Please define the meaning of Ab, as it is repeated twice.

Figure 7: I would suggest moving it to page 8 to be close to related paragraph.

Figure or Fig (lines 159, 252 and 263), please follow MDPI style.

Space/Blank: in few positions, a space is required to separate a word from punctuation, e.g. line 180, 293, 294, 324, 344, etc.

Author Response

Dear Editor.

Thanks for giving me the good comments and your pertinent opinions! I have made appropriate amendments and explanations. Responses to the comments are listed as below:

Point 1: Line 11: albite were experimentally analyzed by density functional theory, and the … I think the authors meant computationally/theoretically instead of experimentally.

Response 1: Thank you very much for your suggestions. I have corrected it to " albite were theoretically analyzed by density functional theory, and the..." in the manuscript.

Point 2: Lines 117: and the structure is shown in Figure 1. Please change to structures are shown ….

Response 2: I have changed the Structure is shown in line 117 to Structures are shown.

Point 3: Line 221:Please define the meaning of Ab, as it is repeated twice.

Response 3: I have defined Ab as an abbreviation for Albite at line 66.

Point 4: Figure 7: I would suggest moving it to page 8 to be close to related paragraph.

Response 4: Thanks for your reminder, Figure 7, has been moved to page 8 to be close to related paragraph.

Point 5: Figure or Fig (lines 159, 252 and 263), please follow MDPI style.

Response 5: Thank you for your prompt, as suggested by MDPI style, I have changed Fig (lines 159, 252 and 263) to Figure2, Figure6 and Figure7.

Point 6: Space/Blank: in few positions, a space is required to separate a word from punctuation, e.g. line 180, 293, 294, 324, 344, etc.

Response 6: I have revised as prompted.

Reviewer 2 Report

In this paper, authors investigate the characteristics of surface active sites of albite by density functional theory, and the activation of the C-H bond of methane of albite catalyst and the reaction mechanism of preparing C2 hydrocarbons by nonoxidative coupling were studied. This work is original and meaningful for the study of catalytic conversion of methane! I recommend it bo be published after authors solving the following problems.

1. Why is the adsorption energy of CH3 and H in Line 177 of Page4 inconsistent with the values in Table 1?

2. Fig.7 has the problem of unclear labeling. In the figure, which graph corresponds to (001) and which graph corresponds to (010)? The energy barrier of CH3 overflowing from (001) to (010) is 306.37 kJ/mol, which is a large energy barrier. Did the authors calculate the energy barrier of CH3 coupling occurring on (001) plane? Is it possible that CH3 does not need to overflow to the (010) surface, and coupling can occur on the (001) surface? By the same token, does the same thing apply to the overflow of H? In addition,Why does the desorption energy of CH3 and H in Figure 7 differ from that in Table 1?

3. How can the authors determine that the dehydrogenation of C2H6 to C2H4 is taking place in the gas phase and not on the catalyst surface?

4. For Figure 7, please add the corresponding structure of each intermediate in the reaction path calculation.

Author Response

Dear Editor.

Thanks for your pertinent opinions! In view of the opinions of the reviewers, I have made appropriate amendments and explanations. Responses to the comments are listed as below:

Point 1: Why is the adsorption energy of CH3 and H in Line 177 of Page4 inconsistent with the values in Table 1?

Response 1: Thanks for your reminder, I have changed the adsorption energy of CH3 and H in line 177 on page 4 to "The adsorption of CH3 and H on the (010) surface is stronger than that on the (001) surface, and their adsorption energies are 150.09kJ/mol and 102.4kJ/mol higher, respectively."

Point 2: Fig.7 has the problem of unclear labeling. In the figure, which graph corresponds to (001) and which graph corresponds to (010)?The energy barrier of CH3 overflowing from (001) to (010) is 306.37 kJ/mol, which is a large energy barrier. Did the authors calculate the energy barrier of CH3 coupling occurring on (001) plane? Is it possible that CH3 does not need to overflow to the (010) surface, and coupling can occur on the (001) surface? By the same token, does the same thing apply to the overflow of H? In addition,Why does the desorption energy of CH3 and H in Figure 7 differ from that in Table 1?

Response 2: (1)I have marked Figure 7 according to modification suggestions, that is, the left side corresponds to (001) and the right side corresponds to (010). (2)As to why CH3 and H are not coupled on the surface of (001), I have described in line 255 on page 7 of the manuscript, that is,” at the same time, because the Lewis acid site Al on the (001) surface has an empty orbital and CH3- has a lone electron pair, after the two bonding, that is, CH3- adsorbed on (001) surface can neither give nor gain electrons, so C-C coupling does not occur on (001) surface. Similarly, the spilled H+ cannot gain electrons from Lewis acid site Al in (001) surface to form H-. Therefore, the coupling reaction of CH3 and H overflowers on (001) surface can only occur on (010) surface,…”.(3)Table 1 shows the "adsorption energy", Figure 7 describes the "desorption energy".

Point 3: How can the authors determine that the dehydrogenation of C2H6 to C2H4 is taking place in the gas phase and not on the catalyst surface?

Response 3: As for your question that " the dehydrogenation of C2H6 to C2H4 is taking place in the gas phase and not on the catalyst surface", it has been confirmed in the previous experimental studies of Ye Chen, Xin Wang et,al. [13], that is, the selectivity of C2 hydrocarbon basically does not change with the temperature between 873K and 1073K. While the selectivity of C2H6 decreases with the increase of temperature, the selectivity of C2H4 increases with the increase of temperature. Therefore, we determine that C2H4 is the product of dehydrogenation of C2H6 in the gas phase. I made a reference annotation at the end of the sentence "C2H4 is the product of dehydrogenation of C2H6 in the gas phase." in line 283 of page 9.

Point 4: For Figure 7, please add the corresponding structure of each intermediate in the reaction path calculation.

Response 4: Thank you for your prompt, I have added the corresponding structures of three intermediates of Figure 7.